# Development of Innovative Vitamin D Enrichment Designs for Two Typical Italian Fresh Cheeses: Burrata and Giuncata

**DOI:** 10.3390/molecules28031049

**Published:** 2023-01-20

**Authors:** Agnese Santanatoglia, Franks Kamgang Nzekoue, Alessandro Alesi, Massimo Ricciutelli, Gianni Sagratini, Xinying Suo, Elisabetta Torregiani, Sauro Vittori, Giovanni Caprioli

**Affiliations:** 1School of Pharmacy, University of Camerino, via Sant’ Agostino 1, 62032 Camerino, Italy; 2Sabelli Group, Basso Marino, 63100 Ascoli Piceno, Italy; 3School of Biosciences and Veterinary Medicine, University of Camerino, via Gentile III da Varano, 62032 Camerino, Italy

**Keywords:** vitamin D_3_, fortification design, cheese, giuncata, burrata, formulation, quality assurance

## Abstract

The aim of this research was to develop innovative cheeses fortified with vitamin D3 (VD3). Formulation studies and analyses of textural properties and chemicals were carried out for these developments. Two traditional Italian varieties of cheese (giuncata and burrata) were studied. For giuncata, the fortification of milk for cheese production provided a VD_3_ retention level of 43.9 ± 0.6% in the food matrix. For burrata, the VD_3_ ingredient was incorporated into the creamy inner part after mixing, maintaining the textural quality of the product (adhesiveness 4.3 ± 0.4 J × 10^−3^; firmness 0.7 ± 0.0 N; and cohesiveness 0.8 ± 0.2). The optimized enrichment designs allowed to obtain homogenous contents of VD_3_ during the production of giuncata (0.48 ± 0.01 µg/g) and burrata cheeses (0.32 ± 0.02 µg/g). Moreover, analyses revealed the high stability of VD_3_ during the storage of the two fortified cheese types (2 weeks, 4 °C). These fortification designs could be implemented at an industrial scale to obtain new cheese types enriched in VD_3_ and thus contribute to the reduction in VD deficiency prevalence.

## 1. Introduction

Vitamin D (VD) deficiency is a major public health issue in the world. Indeed, more than 1 billion people suffer from vitamin D deficiency worldwide, with a high prevalence in the elderly (61–91%), indoor workers (50–80%), and infants (40–60%) [1,2]. Recently, VD deficiency has been correlated in various studies with the severity and mortality of acute respiratory infections (ARIs) such as COVID-19 [3,4,5]. Indeed, the binding to VD receptors by the active form of VD (1,25-dihydroxy vitamin D, 1,25(OH)_2_D) can inhibit the production of pro-inflammatory cytokines (interleukin-6 and tumor necrosis factor-α) by macrophages and prevent T- and B-cell differentiation and proliferation, which are exacerbated in pulmonary inflammatory responses with high risks of respiratory insufficiency [6,7,8]. Therefore, vitamin D supplementation was suggested as an efficient, inexpensive, and safe strategy to improve the vitamin D status of deficient populations in the management of COVID-19 [9]. Indeed, giving an overview of the vitamin D status of European countries, [10] reported that countries with the lowest mean blood levels of VD showed the highest infection and death rates in Europe of COVID-19. Contrarily, northern countries with VD supplementation policies such as Sweden and Finland showed the lowest VD deficiency and similarly the lowest COVID-19 mortality rates. In addition, VD deficiency is also associated with other important health issues, including osteoporosis and fractures in the elderly and autoimmune diseases (diabetes and rheumatoid arthritis) [11]. Considering highly affected Southern European countries such as Spain and Italy, VD supplementation could be considered an adaptable public health strategy to meet the recommended dietary allowance (RDA: 10–15 µg per day) and improve the VD status of at-risk populations [12]. This can be accomplished through a VD food fortification strategy, as carried out in Nordic countries. The VD enrichment of milk and dairy products is reported in various studies [13]. However, prolonged and incorrect consumption of vitamin D supplementation may induce hypercalcemia, hypercalciuria, and hyperphosphatemia, considered to be the initial signs of vitamin D intoxication, so it was not consumed without limitations [14]. Furthermore, considering the cheese matrix, few experiments have been performed on a restricted number of cheese varieties, in particular cheddar cheese [15,16,17], cottage cheese [18], gouda cheese [19], mozzarella cheese [20], or ricotta cheese [21]. The development of a VD enrichment design for cheese application requires the homogenous incorporation and distribution of VD into the cheese matrix without modification of the cheese production steps to preserve its yield and quality. In addition, the success of a fortification model involves the stability of the incorporated ingredient into the final product during its shelf-life. Considering the cheese matrix, VD-enriched milk can be used for the development of enriched cheese. However, enrichment studies reported different rates of retention (40–90%) and stability (80–100%) of VD3 during cheese-making and storage [15,16,17,18,19,22]. Therefore, it is important to develop specific fortification designs and assess the VD stability according to the cheese variety and the fortification level targeted. The constant innovation in functional foods’ development and the necessity of a larger number of enriched dairy products are the key factors to improve the dietary intake of VD in deficient populations. For instance, in Italy, dairy products, which are highly appreciated and commonly consumed, could represent ideal food matrixes for VD enrichment. Therefore, the present study aims to set up VD enrichment systems for typical Italian dairy products. Two traditional fresh cheese varieties were selected to develop and validate the enrichment design, burrata and giuncata, which are typical cow milk cheeses highly consumed in the Italian dietary patterns [23,24]. To our knowledge, despite their high consumption, these cheese varieties have never been considered in VD fortification studies. To develop these new functional dairy products, fortification designs were optimized before validation through a scale-up production and verification of VD stability, distribution, and concentration homogeneity.

## 2. Results

### 2.1. Laboratory-Scale Development of VD_3_ Enrichment Systems 

#### 2.1.1. Enrichment Design for Giuncata Cheese

To determine the RL (recovery level) of VD in cheese, 500 mL of milk was fortified with 100 mg of bioactive powder (2.5 mg of VD_3_/g of powder). The fortified milk was used for cheese-making and the levels of VD_3_ were monitored. From 500 mL of milk, 130 ± 8 g of cheese was obtained, corresponding to a cheese yield of 26 ± 2%. Table 1 reports the levels of VD_3_ in milk, cheese, and whey after fortification. After normal cheese-making, the RL of VD_3_ was 43.9 ± 0.6%. Therefore, to reach a concentration of 50 µg/100 g of giuncata cheese, it is necessary to add 150 µg of VD_3_ in milk and thus 60 mg of bioactive powder. 

#### 2.1.2. Enrichment Design for Burrata Cheese 

Laboratory-scale burrata enrichment started with yoghurt enrichment, blending 1 kg of yoghurt with 0.2 g of bioactive powder to reach a concentration of 50 µg/100 g of yoghurt. After 10 min of blending, six samples were collected and analyzed to assess the homogeneity of VD distribution into the fortified matrix. VD levels ranged between 52.8–55.6 µg/100 g with a mean concentration of 54.2 ± 1.1 µg/100 g. This low variation in concentration (≤2%) indicates a homogenous content of VD in the fortified yoghurt matrix after 10 min of blending. Moreover, the effect of the blending on the textural properties of yoghurt was measured after 5 min and 10 min of treatment. As expected, blending was found to slightly but not significantly alter textural properties (firmness, adhesiveness, and cohesiveness) of yoghurt. Textural attributes of the fortified samples were, therefore, comparable to those of the control yoghurt (Table 2). 

### 2.2. Assessment of the Distribution of the Bioactive Compound into Cheese Matrixes

#### 2.2.1. Distribution of VD_3_ into Giuncata Cheese Matrix

The distribution of VD into the giuncata form is essential to ensure a similar intake from slice-to-slice consumption. It was performed after one week. Table 3 shows the VD levels into the upper, middle, and lower portions of fortified giuncata cheese. No statistically significant difference was observed among the VD levels in the three portions showing an RSD (relative standard deviation) of 1.7%. These results confirmed that inside the giuncata cheese matrix VD is homogeneously distributed after cheese-making. 

#### 2.2.2. Distribution of VD into Burrata Cheese Matrix

Burrata cheese parts were analyzed separately after one week of cheese-making to assess the distribution of VD into the burrata matrix. From this sample, analyses were performed showing a concentration of 0.32 ± 0.02 µg/g (Table 3). No detected levels were observed in the outer casing, showing that VD does not migrate from the inner to the outer portions of the cheese. Moreover, analyses revealed that the highest levels (0.52 ± 0.02 µg/g) remained in the creamy inner part (cream + yoghurt), while a significant transfer of VD was observed in the inner stretch curds (0.15 ± 0.04 µg/g). 

### 2.3. Industrial-Scale Application: Homogeneity of the Production Batch

#### 2.3.1. Giuncata Cheese

The developed enrichment designs were applied on industrial scales for validation. For giuncata cheese, the VD fortification was carried out on 350 kg of pasteurized milk to obtain a VD enrichment level of 50 µg/100 g of cheese. Based on the laboratory experiments (RL = 43.9%) and the cheese yield (26%), from 350 kg of milk approximately 91 kg of cheese was obtained, thus requiring an initial enrichment with 103.7 mg of VD_3._ Therefore, milk maintained at 37 °C was fortified with 41.5 g of bioactive powder (31.7 ± 0.4 µg of VD_3_/100 g of milk), and cheese-making was then performed. The obtained fortified cheese samples were collected and analyzed to confirm the fortification design and assess the homogeneity of the VD content into the production batch. Twelve samples were randomly selected and analyzed at T0, showing a mean concentration of 47.8 ± 0.8 µg/100 g. These levels are close to the targeted fortification level (50 µg/100 g). Moreover, the low variation observed among the analyzed samples (RSD% = 1.7%) confirmed the homogenous content of VD_3_ in the cheese production batch. 

#### 2.3.2. Burrata Cheese

For burrata cheese, the targeted enrichment level was 50 µg/100 g of the inner filling (stracciatella + yoghurt). Using 10 kg of yoghurt, which represents 30% of the inner filling (33.5 kg), 16.75 mg of VD_3_ was added corresponding to 6.7 g of bioactive powder. Thus, 10 kg of yoghurt was mixed for 10 min with the bioactive powder and then combined with the stracciatella (stretch curds and cream) to obtain a homogeneous fortified product. The inner filling was then placed in a dispenser used to fill the outer casing (mozzarella wrapper). Finally, the burrata was closed manually, packaged, and stored. The VD_3_ level was estimated on the whole product and the inner filling, analyzing 12 burrata samples selected randomly (T0). Analyses showed mean concentrations of 48.2 ± 1.0 µg/100 g for the inner filling and 29.6 ± 2.2 µg/100 g for the whole product. Levels resulted lower in the whole product due to the outer casing, which represents 36–42% of the burrata weight. The variation in VD_3_ levels in the fortified burrata cheese is 7.5%, confirming a consistent distribution of VD_3_ in the production batch. However, compared to giuncata cheese, a higher variation was observed in burrata cheese due to the differences in the weight contribution of the outer casing from sample to sample. 

### 2.4. Stability of Vitamin D in Fortified Cheese

Figure 1 shows the stability of VD_3_ in fortified giuncata (B) and burrata (A) cheese during their shelf life. For giuncata (B) cheese, the mean VD_3_ levels were 47.8 ± 0.8 µg/100 g at T0, 48.2 ± 3.3 µg/100 g after one week (T7), and 48.0 ± 0.8 µg/100 g after 2 weeks of storage (T14). Therefore, VD_3_ levels remained stable during giuncata storage, with no statistically significant difference among the time points. In fortified burrata (A) cheese, analyses revealed VD_3_ levels at 25.1 ± 2.5 µg/100 g at T7 and 27.2 ± 2.1 µg/100 g at T14. These levels are not statistically different from those reported after cheese-making (T0), confirming the stability of VD_3_ during the shelf life of giuncata. 

## 3. Discussion

The development of functional cheese enriched in VD_3_ has been reported in a few studies, with experiments mostly performed on cheddar cheese. However, considering the high prevalence of VD deficiency, especially in Southern Europe, it is necessary to develop new VD_3_ enrichment designs for more cheese varieties. Therefore, this study focused for the first time on the enrichment of two traditional fresh cheese varieties highly consumed in Italy: giuncata and burrata cheeses. Therefore, given the amount of fortification obtained from our study, 0.48 ± 0.01 µg/g for giuncata and 0.32 ± 0.02 µg/g for burrata, if they were divided into pieces, a smaller portion of giuncata (20.8–31.25 g) than burrata (31.25–46.9 g) can satisfy the recommended dietary intake (RDA: 10–15 µg per day) and can improve the state of VD_3_ deficiency. The development of cheeses fortified in bioactive ingredients requires the development of an enrichment design able to appropriately incorporate the functional compound into the cheese matrix [25]. Indeed, the enrichment should ensure the proper retention and protection of the bioactive compound in the cheese matrix during the production and shelf life [26]. Our analyses revealed the homogenous distribution and stability of VD_3_ in giuncata and burrata cheese matrixes. Moreover, the industrial application validated the enrichment designs developed in the laboratory. Additionally, the development of a VD_3_-fortified cheese improves its nutritional properties and health benefits. For example, giuncata and burrata, which, like most cheeses, are a good source of protein and calcium, may further improve bone health when enriched with VD_3_ [27]. However, fats, which provide cheese creaminess and contribute to VD_3_ stability, are also associated with cardiovascular diseases (CVD) [28]. Therefore, people following hypolipidemic diets to manage or prevent CVD will be advised to limit the consumption of dairy products with high levels of saturated fats. However, VD_3_-enriched cheese can be eaten as part of a healthy balanced diet [29]. The outbreak of severe acute respiratory syndrome coronavirus 2 (SARS-CoV-2) associated with severe socio-economic costs exposed the fragility of the global healthcare systems [30,31]. Indeed, the actual pandemic statistics suggest that health systems should move forward with more preventive actions by promoting healthy lifestyles and diets to reduce the impact of chronic diseases and future outbreaks of acute respiratory infections (ARIs) [32,33,34,35]. In this context, the development of foods able to provide bioactive compounds such as VD in the organism should be valorized and promoted. 

## 4. Material and Methods

### 4.1. Chemicals and Reagents

HPLC-MS grade methanol and acetonitrile were provided by Carlo Erba (Cedex, France). Ultrapure water was obtained from the Milli-Q SP Reagent Water System (Millipore, Bedford, MA, USA). Hexane was supplied by Carlo Erba (Milan, Italy), while ethanol was supplied by Fisher Scientific (Loughborough, UK). Before use, all the solvents and solutions were filtered through a 0.45 µm filter from Supelco (Bellefonte, PA, USA). Analytical standards of VD_2_ (ergocalciferol, C_28_H_44_O, CAS N° 50-14-6, MW 396.7 g/mol) and VD_3_ (cholecalciferol, C_27_H_44_O, CAS N° 67-97-0, MW 384.6 g/mol) were provided by Sigma Aldrich (Milan, Italy). Stock solutions of 1 mg/mL for each standard were prepared in acetonitrile and stored at 4 °C before adequate dilution in acetonitrile for the daily preparation of the standard working solutions. 

### 4.2. Cheese Samples and Compositional Analyses

Giuncata and burrata cheeses (Figure 2) were produced from cow milk by Sabelli Group (Ascoli Piceno, Italy), a cheese-making company, through standardized industrial procedures. Giuncata is a white and soft fresh cheese obtained from a traditional cheese-making process. Briefly, pasteurized raw milk is heated and maintained at 37 °C. Salt and rennet are added and after curdling, and the curd is cut by the cheesemaker to favor the cheese–whey separation. Then, the curd is collected in perforated baskets for whey draining, obtaining the final curd mass, which can be shaped in cylindrical or cuboidal forms using specific molds. After shaping, the final cheese is kept in cold water (10 °C) for 2 h before confectioning and storage. Burrata is a fresh cheese with a rounded shape made from mozzarella and stracciatella. Indeed, burrata is made up of a small bag of mozzarella paste containing stracciatella, a mixture of stretch curds and cream. In particular, the filling of burrata used in this study was made of a classic stracciatella (70%) and poured yogurt (30%) produced by Caseificio Val d’Aveto SRL (Genoa, Italy). Once produced, burrata and giuncata are packaged in preserving liquid and stored at 4 °C for a total shelf life of 14 days.

The macronutrient compositions of cheeses (giuncata and burrata) are reported in Table 4. The total proteins were determined by the Kjeldahl method, salt analysis was performed by the Volhard method, and lipid levels were assessed by extraction with petroleum ether. Then, the solvent was evaporated to dryness at 95–100 °C and weighed according to the AOAC official method (n° 2011.14 for protein analysis, n° 935.43 for salt, and n° 974.09 for fat) [36]. The total carbohydrates and energy values were calculated by differences.

### 4.3. Design of Vitamin D Enrichment Systems in Cheese: Laboratory-Scale Optimization

The enrichment was performed with a VD_3_ ingredient provided by A.C.E.F Spa (Piacenza, Italy) containing 2.5 mg of VD/g of powder. VD enrichment systems were developed and validated to reach fortification levels of 50 µg/100 g for burrata filling and giuncata.

### 4.4. VD_3_ Enrichment System for Giuncata Cheese

To obtain a fortified giuncata cheese, the VD fortification was performed in the milk before starting the cheese-making procedure. Laboratory-scale production was realized, adding the VD ingredient in 500 mL of milk, and the cheese yield was determined.
Cheese yield = (quantity of milk/quantity of obtained cheese) × 100(1)

Various trials were attempted through laboratory-scale cheese manufacture models to determine the cheese retention of VD and estimate the initial fortification levels to reach the targeted levels in the final products. Concentrations and quantities of VD in the whey were also determined. The recovery level (RL) of vitamin D in giuncata cheese was calculated according to the following equations [21]: RL (%) = [Total VD_3_ in cheese (µg)/Total VD_3_ introduced in the milk (µg)] × 100 (2)
Total VD_3_ in cheese (µg) = [concentration of VD_3_ in cheese (µg/g) × quantity of cheese (g)](3)

Determining the RL, it was possible to establish the fortification levels in milk to reach 50 µg/100 g in cheese. The optimized conditions were then applied at the industrial scale to validate the developed design.

### 4.5. VD_3_ Enrichment System for Burrata Cheese

The burrata was fortified into the yoghurt before mixing with stracciatella. It was decided to add VD to yoghurt because in this way it is possible to obtain a homogeneous system. After weighing the VD ingredient, it was added to 1 kg of yoghurt and then mixed slowly for ten minutes using a screw mixer (KitchenAid, Milan, Italy). The homogeneity of VD distribution in yoghurt was evaluated. Moreover, the effect of mixing on the textural quality of the yoghurt was also assessed. Textural properties were measured using a Food Texture Analyzer (TA1 Texture Analyzer, AMETEK, Berwin, PA, USA) equipped with a 100N load cell. Firmness (peak force of the first compression cycle, N), adhesiveness (work conducted between the end of the first compression and the beginning of the second compression, J), and cohesiveness (ratio of work of the second and first compression cycles) were evaluated as described by [37]. A two-cycle compression was applied using a 25 mm stainless steel diameter cylinder probe at a speed of 1 mm/s to a sample depth of 30 mm. The optimized conditions were then applied to the industrial scale to validate the fortification systems. 

### 4.6. Determination of Vitamin D through HPLC Analyses

VD_3_ analysis was performed by HPLC-DAD (1260 Infinity, Agilent Technologies, CA, USA) using a Gemini C18 column (250 × 3.0 mm, 5 µm, Phenomenex, Torrance, CA, USA) at a controlled temperature of 40 °C. The analyte separation was performed in gradient mode with water (A) and methanol (B) as the mobile phase: 0–5 min, 80% of B; 5–7 min, 100% of B; 7–17 min, 100% of B; 17–18 min, 80% of B; and 18–20 min 80% of B. The injection volume was 20 µL and analytes were quantified at 265 nm as detection wavelength. VD_3_ was the monitored analyte and VD_2_ was used as the internal standard (I.S). The extraction of VD_3_ from fortified matrices (milk and yoghurt), final cheese, and remaining whey was performed by hot saponification followed by extraction with hexane [20]. Briefly, to 5 g of sample, 200 µL of I.S solution (10 µg/mL) was added to samples to reach a final concentration of 2 µg/mL. Then, 12 mL of ethanol and 4 mL of KOH solution (1 g/mL) were added, and saponification was conducted at 65 °C for 45 min under constant agitation. After saponification, samples were cooled in ice and 15 mL of NaCl solution (1 g/100 mL) was added. VD extraction was performed with hexane three times (10 mL × 3), and the collected extracts were dried using a rotavapor and reconstituted with 1 mL of acetonitrile. The final extract was filtered using a 0.45 µm filter before HPLC analyses. 

### 4.7. Homogeneity Study of Production Batches and Assessment of the Distribution of the Bioactive Compound into Cheese Matrixes

The distribution of VD_3_ after enrichment was assessed in giuncata and burrata cheese matrixes. For giuncata, cheese slices were divided into three portions: upper, middle, and lower portions (Figure 2A). Each portion was analyzed and their difference in terms of VD_3_ content was evaluated. For burrata, the levels of VD_3_ were determined in the three different parts of cheese: the outer casing (bag of mozzarella paste), the stretch curds (internal filling), and the creamy inner part (cream + yoghurt) (Figure 2B). The concentration in whole cheese was determined by milling the whole burrata into a mixer grinder, thus obtaining a semi-solid sample. After the industrial production of fortified burrata and giuncata cheeses, the homogeneity of the production batches was evaluated by analyzing the VD_3_ content in 12 samples randomly selected for each kind of cheese. The variation in VD_3_ content from cheese creation to confection was thus assessed, and the mean concentrations were used as T0 levels. 

### 4.8. Stability of Vitamin D in Fortified Cheeses

The chemical stability of VD_3_ inside the fortified cheese matrixes (giuncata and burrata) was evaluated throughout the shelf life based on the measurement of VD_3_ content in cheese during 2 weeks of storage at 4 °C. Analyses were performed after one week (T7) and two weeks of storage (T14). At each time point, 6 replicates from 6 forms of cheese randomly selected were analyzed. The obtained levels were compared with VD_3_ content at T0. 

### 4.9. Statistical Analyses

All the analyses were performed with more than 3 replicates (*n* ≥ 3), and data are expressed as mean ± standard deviation (S.D). Relative standard deviation (RSD%) was used to assess the uniformity of VD content in analyzed samples.
RSD% = (S.D/mean) × 100 (4)

RSD% ≤ 10% indicated a homogenous content. The one-way analysis of variance (ANOVA) was used to perform statistical analysis, and differences between means were considered statistically significant with *p* < 0.05. 

## 5. Conclusions

Giuncata and burrata cheeses were successfully enriched in VD_3_ through the enrichment designs developed and validated in this study. Analyses revealed that giuncata and burrata cheeses are suitable dairy matrixes for VD_3_ fortification. Indeed, the enrichment designs allowed us to obtain fortified cheeses with homogenous and stable content of VD_3_ during production and storage (2 weeks at 4 °C). Developing innovative functional dairy products fortified with VD3 could improve the VD status in deficient populations. Therefore, these designs can be applied at industrial scales for functional cheese production.

## Figures and Tables

**Figure 1 molecules-28-01049-f001:**
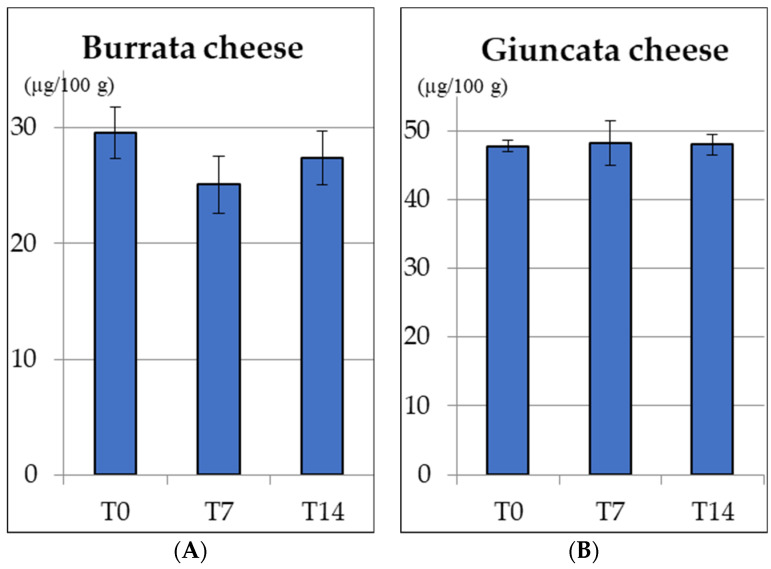
Stability of vitamin D_3_ during storage burrata and giuncata cheeses (14 days at 4 °C). No statistical difference between the levels was found over the storage (*p* < 0.05).

**Figure 2 molecules-28-01049-f002:**
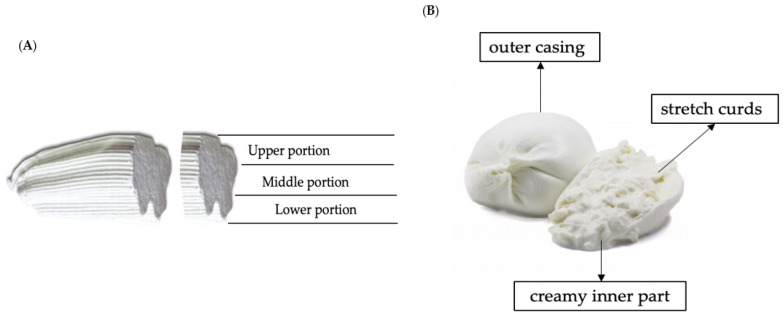
Italian cheeses studied for vitamin D_3_ enrichment: (**A**) giuncata cheese; (**B**) burrata cheese.

**Table 1 molecules-28-01049-t001:** Distribution of VD_3_ during giuncata production.

	Quantity (g)	Concentrations (µg/g)	Total Quantity of VD_3_ (µg)	Proportions (%)
Milk	500	0.45 ± 0.0	225 ± 0.5	100
Cheese	130	0.76 ± 0.01	98.8 ± 1.3	43.9 ± 0.6
Whey	370	0.34 ± 0.01	125.8 ± 3.7	55.9 ± 1.6

**Table 2 molecules-28-01049-t002:** Effect of blending on the textural characteristics of fortified yoghurt.

	Adhesiveness (J)	Firmness (N)	Cohesiveness
Control yogurt	5.3 ± 1.1 J	0.8 ± 0.1 N	0.8 ± 0.0
5 min blending	4.5 ± 0.5 J	0.7 ± 0.0 N	0.8 ± 0.1
10 min blending	4.3 ± 0.4 J	0.7 ± 0.0 N	0.8 ± 0.2

**Table 3 molecules-28-01049-t003:** Distribution of vitamin D_3_ into cheese matrix.

Giuncata Cheese	Upper Part	Middle Part	Lower Part	Whole Cheese	Units
	0.48 ± 0.01	0.49 ± 0.01	0.48 ± 0.00	0.48 ± 0.01	µg/g
**Burrata Cheese**	**Outer Casing**	**Stretch Curds**	**Creamy Inner Part**	**Whole Cheese**	**Units**
	0.0 ± 0.0 ^a^	0.15 ± 0.04 ^b^	0.52 ± 0.02 ^c^	0.32 ± 0.02	µg/g

^a,b,c^ Values with different upper-case letters are significantly different (*p* < 0.05).

**Table 4 molecules-28-01049-t004:** Nutrition facts of burrata and giuncata cheese.

	Burrata	Giuncata
Energy value	239 ± 9 Kcal	186 ± 6 Kcal
Fats	21.2 ± 0.8 g	15.0 ± 0.6 g
of which saturated fats	15 ± 0.6 g	9.8 ± 0.4 g
Carbohydrates	1.5 ± 0.0 g	1.8 ± 0.1 g
of which sugar	0.7 ± 0.0 g	0.6 ± 0.0 g
Proteins	12.3 ± 0.4 g	11.2 ± 0.2 g
Salt	0.3 ± 0.0 g	0.7 ± 0.0 g

## Data Availability

Not applicable.

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
