# Peer review of "Development of Innovative Vitamin D Enrichment Designs for Two Typical Italian Fresh Cheeses: Burrata and Giuncata"

_molecules, 2023, doi:10.3390/molecules28031049_

Round 1
Reviewer 1 Report (Previous Reviewer 3)
Authors have improved the manuscript. If it has been accepted for special issue I have no further comments.
Author Response
REFEREE 1
Authors have improved the manuscript. If it has been accepted for special issue, I have no further comments.
R: Thank you.
Reviewer 2 Report (Previous Reviewer 1)
Most of the comments in the review have been taken into account. However, not all of them. Please refer to my comments below.
L. 92-94: The amount of milk this supplement applies to is not specified. In the case of 500 mL of milk, this will be 60 mg of powder. Not 45.6 mg. Please correct this sentence.
L. 126-127: In my opinion, this sentence should be moved to the Materials abd methods section.
L. 169-170: The mistake has not been corrected - "(…) were 47.8 ± 0.8 μg/100 g at T0, 48.2 ± 3.3 μg/100 g after one week (T7), and 48.0 ± 0.8 μg/100 g after (…)"
L. 184: There is "gg", please correct to "g". The calculation is correct.
References: Please remove the comments written in Italian (ref. no. 10 and 26).
The reference no. 37 cannot be found on-line. Does it come from a manual used in a company? Please refer to the method that can be reused by other scientists or describe the method.
Author Response
REFEREE 2
Most of the comments in the review have been taken into account. However, not all of them. Please refer to my comments below.
- 92-94: The amount of milk this supplement applies to is not specified. In the case of 500 mL of milk, this will be 60 mg of powder. Not 45.6 mg. Please correct this sentence.
R: According to referee’s suggestion, the suggested change was made (lines 91-92).
- 126-127: In my opinion, this sentence should be moved to the Materials and methods section.
R: According to referee’s suggestion, the suggested change was made (lines 121-123). The sentence has been moved to the “Materials and Methods” section (lines 305-306).
- 169-170: The mistake has not been corrected - "(…) were 47.8 ± 0.8 μg/100 g at T0, 48.2 ± 3.3 μg/100 g after one week (T7), and 48.0 ± 0.8 μg/100 g after (…)"
R: According to referee’s suggestion, the suggested change was made (line 166). We thank the reviewer for bringing the point to our attention.
- 184: There is "gg", please correct to "g". The calculation is correct.
R: According to referee’s suggestion, the suggested change was made (line 180).
References: Please remove the comments written in Italian (ref. no. 10 and 26).
R: According to referee’s suggestion, the suggested change was made (lines 375/425). We thank the reviewer for bringing the point to our attention.
The reference no. 37 cannot be found on-line. Does it come from a manual used in a company?
Please refer to the method that can be reused by other scientists or describe the method.
R: Following Reviewer 2 comment, reference n. 37 was now reported in the right form (lines 276/459). We thank the reviewer for bringing the point to our attention.
Reviewer 3 Report (Previous Reviewer 2)
Dear authors,
I revised your manuscript and concluded that all the required changes and improvements were done.
Author Response
REFEREE 3
I revised your manuscript and concluded that all the required changes and improvements were done.
R: Thank you.
Round 2
Reviewer 2 Report (Previous Reviewer 1)
Thank you for the answers. I have only one comment. Only one part of the sentence in lines 89-91 has been corrected. In my opinion, it should be stated like this:
Therefore, to reach a concentration of 50 μg/100 g of giuncata cheese, it is necessary to add 150 μg of VD3 in 500 mL milk and thus, 60 mg of bioactive powder.
Author Response
According to referee’s suggestion, the suggested change was made (lines 89-90).
This manuscript is a resubmission of an earlier submission. The following is a list of the peer review reports and author responses from that submission.
Round 1
Reviewer 1 Report
The manuscript is interesting and its content is significant. Please refer to my comments below.
English language needs editing. There is a mistake even in the title.
Please state clearly the aim of the study in the abstract.
Introduction: Please raise the risk of vitamin D overdose. This vitamin cannot be consumed without limitations.
Results:
L. 80. unit for the amount of the powder is wrong
L. 85-86. Please check the calculation. When you put 100 mg of the powder into the milk you get 76ug vit. D / 100 g of cheese. Therefore, to obtain 50 ug/100 g you need more than 45.6 mg of the powder per 500 mL of milk.
Please explain why there should be 50 ug of vit. D per 100 g of cheese or yoghurt. On what basis was it chosen?
L. 97, 112, 153. Please cite references to explain why less than 2% proves homogenous distribution and why the vit. D variation of 7,5% means the consistent distribution.
Table 3. When was distribution of vit. D determined in Giuncata and why was it determined after one week in Burrata? Why not after 0, 7 and 14 days?
L. 119-120. This is the method. Please move to the next section.
L. 132. Please check the calculation.
L. 163. 48.2?
Discussion:
L. 188. The word "normal" is inappropriate here. Please rewrite.
Please discuss, what daily portion of these cheeses will be beneficial for the consumer. Please note the RDA.
Why was this level of supplement chosen? Why was it added to the yoghurt?
Material and methods:
L. 230-234. Please give the number of methods from AOAC. How saturated fats, sugar and salt were determined?
L. 261. Please explain how the homogeneity of VD distribution evaluated. Cite references.
L. 309-316. Please explain which results were analyzed using ANOVA and which using Student T-test.
References:
Please provide full bibliographic details for the following references: 34, 35 and 36.
Reviewer 2 Report
Dear authors, you can find my suggestions below:
Line 17. What is the meaning of “the enrichment design into the milk”. Is it “milk enrichment”? Please clarify the sentence.
Line 50-51. Please combine the words in a continuous line.
Line 80. What is RL? Please use its long form for the first time. What is 100 m?
Line 110. What is RSD? Please use its long form for the first time. Please revise this expression: RSD % of 1.7 % or RSD % of 1.7 or RSD of 1.7%.
Line 119. “The concentration into the whole cheese”. It is probably “in whole cheese”
Line 132. What is AN? Please use its long form for the first time.
Discussion is not properly structured. It needed to be revised.
Reviewer 3 Report
Manuscript entitled "Development of Innovative Vitamin D-enrichment Designs for Two Typical Italian Fresh Cheese: Burrata and Giuncata Cheese" describes the well-known topic of vitamin D enrichment of cheeses. The methods used, the results obtained and their presentation are at a very basic level. The term functional food should not be used in the context of fortified products until such properties have been proven in human clinical trials. Any suggestion of this kind for the fortified cheeses described in the manuscript is clearly premature at this stage.
In my opinion, the manuscript also doesn't quite fit the aims and scopes of Molecules. Rather, I suggest a journal typically dealing with food technology.